# Combining Recent Nutritional Data with Prospective Cohorts to Quantify the Impact of Modern Dietary Patterns on Disability–Adjusted Life Years: A Feasibility Study

**DOI:** 10.3390/nu12030833

**Published:** 2020-03-20

**Authors:** Jean-Philippe Krieger, Giulia Pestoni, Anita Frehner, Christian Schader, David Faeh, Sabine Rohrmann

**Affiliations:** 1Division of Chronic Disease Epidemiology, Epidemiology, Biostatistics and Prevention Institute, University of Zurich, Hirschengraben 84, 8001 Zurich, Switzerland; jean-philippe.krieger@gu.se (J.-P.K.); giulia.pestoni@uzh.ch (G.P.); david.faeh@bfh.ch (D.F.); 2Research Institute of Organic Agriculture (FiBL), Ackerstrasse 113, 5070 Frick, Switzerland; anita.frehner@fibl.org (A.F.); christian.schader@fibl.org (C.S.); 3Wageningen University & Research, Animal Production Systems Group, De Elst 1, 6708 Wageningen, The Netherlands; 4Health Division, Nutrition and Dietetics, Bern University of Applied Sciences, Falkenplatz 24, 3012 Bern, Switzerland

**Keywords:** dietary patterns, DALYs, menuCH

## Abstract

Unhealthy diets are commonly associated with increased disability-adjusted life years (DALYs) from noncommunicable diseases. The association between DALYs and dietary patterns can be quantified with individual longitudinal data. This assessment, however, is often based on dietary data collected once at cohort entry, therefore reflecting the impact of “old” dietary habits on morbidity and mortality. To overcome this limitation, we tested the association of contemporary diets with DALYs. First, we defined contemporary dietary patterns consumed in Switzerland with the national nutrition survey menuCH (2014–2015). Second, we identified individuals who consumed similar diets in the NRP–MONICA census-linked cohort (1977–2015). In this cohort, individual data on disease and mortality were used to calculate the DALYs-dietary patterns association using a mixed regression model. A total of 58,771 DALYs from NCDs were recorded in a mean follow-up time of 25.5 years. After multivariable adjustments, the “Swiss traditional” pattern was not associated with an increase in DALYs compared to the “Prudent” pattern. However, individuals following a “Western” pattern had, on average 0.29 DALYs (95% CI 0.02, 0.56) more than those following a “Prudent” pattern, equating to a loss of healthy life of more than three months. These data highlight the feasibility of quantifying the impact of contemporary diets on DALYs without the establishment of new cohorts or the use of nationally aggregated data.

## 1. Introduction

Unhealthy dietary patterns are a leading cause of morbidity and mortality from noncommunicable diseases (NCDs) across the world [1]. Specifically, low consumption of fruits, vegetables, nuts, and high intake of salt are the dietary factors most associated with the global burden of NCDs [2].

Disability-adjusted life years (DALYs), a measure developed by the World Health Organization and the World Bank [3,4], allow for the quantification of the total burden of disease at the population level.

DALYs are classically calculated on a population level, based on aggregated incidence and mortality data [1,3,5]. The use of individual data from prospective studies has been proposed to compute DALYs based on real-life follow-up data rather than modeled population estimates [6,7,8]. Thus, the use of this method in well-chosen cohort studies can provide reliable time- and country-specific estimates of the association between DALYs and dietary patterns.

In most longitudinal studies, however, dietary assessment occurs once at study entry (i.e., potentially decades before the end of the study); thus, the DALYs-dietary patterns association calculated based on this information reflects the impact of “old” dietary habits on morbidity and mortality [9]. Although this information is extremely valuable from a public health perspective, the impact of currently consumed diets on DALYs may be more relevant to improve and dynamically adapt nutritional guidelines. This is particularly relevant, as data show considerable shifts in diets over time periods as short as 20 years [10].

To overcome this limitation, we combined two chronologically distinct data sources in order to calculate the association between dietary patterns currently consumed and DALYs from NCDs. In this method, a recent cross-sectional study was first used to characterize the currently consumed dietary patterns [11,12,13], whereas a prospective cohort study was used to quantify the dietary patterns-DALYs association for individuals who consumed a diet similar to the modern dietary patterns [9]. This approach extends previous work on the estimation of DALYs in cohort studies to the calculation of the association between current dietary patterns and DALYs.

## 2. Methods

### 2.1. Study Populations

The cross-sectional population-based survey menuCH was the first national nutrition survey in Switzerland. It was conducted between January 2014 and February 2015 in ten study centers across Switzerland. Swiss residents aged 18 to 75 years old were drawn from a random stratified sample [13,14]. The survey protocol was approved by the lead ethics committee in Lausanne (Protocol 26/13) and by the corresponding regional ethics committees. The survey was registered at the ISRCTN registry under the number 16778734.

The NRP–MONICA census-linked cohort (NRP–MONICA) consists of 2 population-based studies, the National research program (NRP) 1A [15] and the Monitoring of trends and determinants in cardiovascular disease (MONICA) [16], linked with the Swiss national cohort (SNC) [17]. All studies comprised a self-administered simplified 24-hour diet recall (24HDR) checklist. Vital status and cause of death of the NRP1A and MONICA participants were followed through the SNC, a national longitudinal research platform linking census records with federal death and migration records, covering all residents of Switzerland [17,18,19]. SNC and its linkage with MONICA and NRP1A were approved by the Ethics Committee of the Canton of Zurich (KEK-StV no. 13/06 and amendment of 12 June 2008). Secondary analysis of the mortality data of this cohort was registered at the ISRCTN registry under the number 10294463. Procedures of both studies followed the guidelines laid down in the Declaration of Helsinki.

### 2.2. Dietary Assessment

In menuCH, habitual dietary intake was estimated with two 24HDR, one face-to-face, and a second one, two to six weeks later on the phone [12]. The software GloboDiet^®^ (formerly EPIC-Soft^®^, version CH-2016.4.10, International Agency for Research on Cancer (IARC), Lyon, France) [20,21], adapted to Switzerland (GloboDiet^®^ trilingual databases dated 12 December, 2016; IARC, Lyon, France, and Federal Food Safety and Veterinary Office, Bern, Switzerland) was used to record dietary intakes. A sample of 2057 participants completed two 24HDR, and their dietary data were used to derive a posteriori dietary patterns [22]. Multiple factorial analysis and hierarchical clustering were applied on the energy-standardized daily consumption of 17 food categories, and four dietary patterns were identified (“Prudent” *n* = 486; “Swiss traditional” *n* = 744; “Western 1–soft drinks and meat” *n* = 383; “Western 2–alcohol, meat and starchy” *n* = 444) [22].

NRP–MONICA participants were invited to attend a health examination in which they answered a self-administered questionnaire. The 24HDR checklist comprised yes/no questions on meat (white and red), sausage, fish, salad, vegetables, fruit, wholegrain products in the form of dark bread, chocolate, eggs, cheese, milk (as a drink), and yogurt. Alcohol consumption was collected differently in NRP and MONICA and dichotomized as yes (NRP, women ≥20 g/day, men ≥40 g/day; MONICA, drinking spirits or more than one sort of alcohol on the previous day) or no (NRP, women <20 g/day, men <40 g/day; MONICA, not drinking or drinking either wine, beer, or cider). Only participants with complete dietary information were included in the present analysis (*n* = 15,843 out of 17,861; 88.7%).

### 2.3. Adaptation and Dichotomization of MenuCH Dietary Patterns

To assign the diets of NRP–MONICA participants to one of the four previously defined menuCH dietary patterns, we relied on common dietary information between the two studies. However, these studies differed in dietary data quality, such that NRP–MONICA recorded fewer food groups than menuCH (12 vs. 17) and collected only dichotomized information (vs. quantitative information). Therefore, the food information originally used to define menuCH dietary patterns was adapted to match the information available in NRP–MONICA.

First, when menuCH food categories originally used by Krieger et al. [22] did not directly match the available NRP–MONICA food categories, we used menuCH food subcategories (defined in GloboDiet^®^; Appendix A). Second, we dichotomized the four menuCH dietary patterns on the basis of the z-scores of energy-standardized food consumption in the menuCH survey; all z-scores >0.1 were transformed into yes, others in no. When no z-scores were >0.1 for a food category, only the highest z-score was transformed into yes. The results of the dichotomization are shown in Appendix A. Finally, because the patterns “Western 1–soft drinks and meat” and “Western 2–alcohol, meat and starchy” were highly similar after dichotomization, we decided to group them into one “Western” pattern (Appendix A). Therefore, the three patterns considered for this study were: “Prudent” (characterized by the consumption of fish, salad, fruits, vegetables, and eggs), “Swiss traditional” (characterized by the consumption of cheese, milk, yogurt, and chocolate) and “Western” (characterized by the intake of meat, sausage, and alcohol).

### 2.4. Assignment of Diets of NRP–MONICA Participants to A Dietary Pattern

The Jaccard similarity index was computed between the diets of each NRP–MONICA participant and the three dichotomized dietary patterns from the menuCH study. Each NRP–MONICA participant was assigned to the dietary pattern for which the Jaccard similarity index was the highest. When Jaccard coefficients were <0.3 for all patterns, NRP–MONICA participants’ diets were considered too dissimilar to the three dietary patterns to be assigned to one, and therefore, were excluded from the analyses. The degree of similarity between the diets of NRP–MONICA participants and the three dichotomized menuCH patterns is shown in Appendix A.

### 2.5. Anthropometric, Sociodemographic and Lifestyle Variables

In NRP–MONICA studies, age, sex, nationality (Swiss, foreign), education (mandatory, upper secondary, tertiary), measured body mass index (BMI, continuous), physical activity (less than once a week, vs. once a week, vs. more than once a week), smoking status (never, former, light (<20 cigarettes/day); heavy smoker (≥20 cigarettes/day)) were assessed. Missing values (15 in BMI, 20 in education, 243 in physical activity, 25 in smoking status) were imputed by linear regression (BMI) or proportional odds models (other ordered variables), using the other available anthropometric, sociodemographic and lifestyle variables as predictors.

### 2.6. Endpoint Assessment

Vital status of participants and the cause of death was available through the SNC. Causes of death were coded according to the International Classification of Diseases (ICD; eighth revision until 1994, tenth revision since 1995).

### 2.7. Calculation of DALYs from NCDs

We calculated DALYs for individuals whose death was attributed to NCDs in the NRP–MONICA cohort. We considered the following causes of deaths as indicative of NCDs: all neoplasms (ICD-8, 140–239; ICD-10, C00–D48), diseases of the circulatory system (ICD-8, 390–459; ICD-10, I00–I99), diseases of the respiratory system (ICD-8, 460–519; ICD-10, J30–J98), diabetes mellitus (ICD-8, 249–250; ICD-10, E10–E14) and chronic liver diseases (ICD-8, 570–573; ICD-10, K70 and K74).

DALYs were computed as the sum of YLL and YLD [5]. YLL is the number of years death occurred earlier than expected. Age, sex, and calendar year-specific life expectancy tables were obtained from the Federal Statistical Office of Switzerland [23]. YLD is the number of years a person lived with a disability, taken into account the severity of this disability (disease duration x disability weight). Disability weights ranged between 0 (no burden) and 1 (death) and were derived from the Global Burden of Disease 2016 disability weights [8,24]. As no date of disease occurrence was provided in the NRP–MONICA cohort, we used disease-specific mean disease duration from the EPIC-NL cohort [6]. A list of disability weights and disease durations used in our analysis is shown in Appendix A. When the disease duration was greater than a participant’s total follow-up time, we used follow-up time as an estimate of disease duration.

### 2.8. Association between Dietary Patterns and DALYs

Because a large number of NRP–MONICA participants have 0 DALY from NCDs at the end of follow-up, the DALYs variable exhibits a peak at 0, followed by a normal distribution. We, therefore, used a two-part model, as described in Struijk et al. [6]. We combined the predicted probability that DALYs >0 from logistic regression (DALYs = 0 or DALYs >0) with the number of DALYs estimated from a linear regression among participants with DALYs >0. Confidence intervals were constructed with bootstrapping of 500 samples. A crude model was adjusted only for age and sex, while a multi-adjusted model was adjusted for age, sex, nationality, education, BMI, physical activity, and smoking status. Individuals nonassigned to a dietary pattern were not included in the analysis. All descriptive and statistical analyses were conducted with R (version 3.5.0) and R-studio (version 1.1.453 for Mac).

### 2.9. Sensitivity Analyses

First, we recalculated the estimates of the DALYs-dietary patterns association, considering the individuals nonassigned to a dietary pattern as a separate category. Second, because the assignment of NRP–MONICA participants to a menuCH dietary pattern depends on an arbitrary cut-off of the Jaccard index (Jaccard >0.3), we tested the effect of lowering or increasing this cut-off on the number of participants assigned to each cluster and on the estimates of the DALYs-dietary patterns association. Finally, to model the uncertainty around participants’ disease durations, we recalculated the estimates of the DALYs-dietary patterns association, introducing a random individual variation in disease duration ranging from ± 10% to ± 50% of the values used in the main analysis. To reduce computation time, bootstrapping of confidence intervals was done with 250 samples (rather than 500) in these two last sensitivity analyses.

## 3. Results

The study included 15,843 participants with a mean follow-up time of 25.5 years (SD 9.1 years). A majority of the participants were Swiss citizens (81.4%), received at least upper secondary education (65.4%), and were neither overweight nor obese (55.4%; Table 1).

Based on their self-reported dichotomized dietary information, participants were assigned to the most similar of the three dichotomized dietary patterns identified in the menuCH study (Appendix A). 6601 (41.7%) participants were assigned to a “Prudent” pattern, 2849 (18.0%) to a “Swiss traditional” pattern, and 2508 (15.8%) to a “Western” pattern (Table 1). The diets of 3885 participants (24.5%) were considered as too dissimilar (all three Jaccard indices <0.3) to be assigned to a pattern. Participants assigned to the “Western” pattern showed the highest percentage of males (71.7%), overweight or obese individuals (51.3%), low physically active individuals (61.2%), and heavy smokers (26.1%) of all groups (Table 1).

A total of 4590 deaths occurred during the study follow-up, among which 3378 deaths were caused by NCDs, representing a total of 58,771 DALYs from NCDs (Table 2). The vast majority of DALYs were attributable to cancers and cardiovascular diseases (Table 2).

In both the crude and multivariable-adjusted two-part model, the “Swiss traditional” pattern was not associated with an increase in DALYs compared to the “Prudent” pattern (Table 3). In the crude model, the “Western” pattern was associated with an increase of 0.55 DALYs (95% CI 0.24; 0.86) compared to the “Prudent” pattern (Table 3). After multivariable adjustments, individuals following a “Western” pattern had, on average, 0.29 DALYs (95% CI 0.02; 0.56) more than those following a “Prudent” pattern, equating to a loss of healthy life of more than three months (Table 3).

When participants nonassigned to a dietary pattern were considered as a separate category in the two-part model, the “nonassigned” group was associated with an increase in 0.27 DALYs (95% CI 0.03; 0.50) compared to the “Prudent” pattern in the crude model (Appendix A). After multivariable adjustments, however, there was no increase in DALYs compared to the “Prudent” pattern (0.15 (−0.10; 0.38)). None of the estimates associated with the “Swiss traditional” or the “Western” patterns were noticeably modified.

The Jaccard index threshold had, as expected, a large influence on the number of participants assigned to the three dietary patterns: the cut-off of 0.3 chosen for this analysis, however, allowed for disregarding the most dissimilar diets, while preserving large group sizes (Appendix A). In fact, crude and multi-adjusted estimates of the DALY-dietary patterns association remained highly similar when the Jaccard index threshold varied from 0 to 0.4 (Appendix A) and became imprecise (wider confidence intervals) above 0.4, due to the decrease in group sizes.

Finally, because the occurrence of diseases was not assessed at the individual level in the NRP–MONICA cohort, we tested the robustness of our results to individual variations in disease duration. Introducing a random individual variation of disease duration up to ± 50% did not noticeably affect the crude and multi-adjusted estimates of the DALY-dietary patterns association (Appendix A).

## 4. Discussion

Based on data of a Swiss census-linked cohort, the intake of a “Western” dietary pattern was associated with a loss of healthy life of more than three months, compared to adherence to a “Prudent” pattern. Similarly, in Dutch cohorts, the a posteriori defined “Prudent” dietary pattern was associated with lower DALYs than the “Western” pattern [7]. Also, a priori diet quality score, such as the Mediterranean diet score [8], or the healthy diet indicator [7], reduced DALYs with increased diet quality. The order of magnitude of changes in DALYs between “Prudent” and “Western” dietary patterns, or between high and low diet quality, are consistent in studies using the same methods in other cohorts [6,7,8].

The first distinctive feature of our study was that we investigated the association between dietary patterns and DALYs from several NCDs using individual data from a large cohort with a long follow-up, extending the previous work with Dutch cohorts [6,7,8]. Compared with aggregated data from burden-of-disease studies, estimating DALYs from cohort studies does not require assumptions on exposure distribution, event rates, and confounding factors. However, this approach requires complete, high-quality individual information (for example, on the time of disease occurrence) that is rarely available in cohort studies. In our cohort, individual time and cause of death, but not the time of disease incidence, were available [15,17,18]. To cope with this lack of individual data, we used disease-specific mean duration from the EPIC-NL cohort [7]; this workaround, however, misses individuals who suffer from NCDs but did not die during follow-up, and those who suffered from NCDs but whose cause of deaths were not attributed to their NCDs. This potentially led to an underestimation of YLD. Conversely, for individuals whose first occurrence of an NCD is fatal (i.e., stroke), the use of the population mean of disease duration potentially led to an over-estimation of YLD. Our sensitivity analysis, however, indicated that the DALYs-dietary patterns association is robust to uncertainty on YLD, mostly because YLD is a smaller contributor to overall DALYs than YLL is.

The second distinctive feature of our study is that we attempted to assess DALYs associated with recently defined dietary patterns, rather than those defined at cohort entry. This approach was motivated by the availability of a nation-wide nutrition survey allowing for the definition of dietary patterns in the Swiss population [11,22,25] and evidence of shifts in dietary patterns over 20 years in Switzerland [10]. For this, we combined two chronologically distinct data sources, i.e., the recent cross-sectional study menuCH to characterize the currently-consumed dietary patterns [22,25] and the census-linked cohort study NRP–MONICA, to characterize participants who consumed a diet similar to the currently-consumed dietary patterns. The main strength of our approach is to investigate the health impacts of modern dietary patterns by looking at the DALYs associated with individuals who consumed similar diets in the past. This approach, however, requires assigning participants of a cohort study to dietary patterns that were defined based on another study. Here, we used the Jaccard index to assess the degree of similarity between diets consumed by the cohort participants and the three dietary patterns defined in the menuCH study. However, because diets evolve, it is expected that diets of certain cohort participants have a low similarity with any of the modern Swiss dietary patterns. Thus, we used a Jaccard index threshold under which cohort participants were not assigned to any modern dietary patterns. The choice of this threshold, however, leads to a trade-off between the number of cohort participants assigned to a pattern and the precision of this assignment, which was quantitatively characterized and illustrated in our sensitivity analyses (Appendix A).

Our study has limitations inherent to the collection of dietary data with 24HDR. 24HDR cannot reliably represent habitual intake at the individual level. This may lead to the misclassification of cohort participants and an underestimation of the DALYs-dietary patterns associations. Also, modern Swiss dietary patterns were quantitatively defined in menuCH [22], whereas only dichotomized dietary data were available from NRP–MONICA. Thus, NRP–MONICA participants whose diets have a high degree of similarity with a Swiss modern dietary pattern (i.e., a high Jaccard index) might still show deviation from the pattern in the quantitative consumption of certain food groups. More generally, we acknowledged that our proposed method assumes that dietary assessment was conducted in a comparable and reliable manner over time, which may not be true in all contexts.

In conclusion, the application of our new method to quantify the impact of modern Swiss diets on DALYs indicates that adherence to “Prudent” or “Swiss traditional” dietary patterns are associated with a reduction in future NCDs burden compared to a “Western” diet. These results are plausible in that they confirm those of the Global Burden of Disease studies indicating that unhealthy dietary patterns are a leading cause of morbidity and mortality from NCDs across the world [1]. The generalizability of this method to other cohort studies needs to be further tested.

## Figures and Tables

**Table 1 nutrients-12-00833-t001:** The characteristics of the NRP–MONICA cohort participants by dietary pattern.

	Overall	Prudent	Swiss Traditional	Western	Nonassigned
	*n* = 15,843	*n* = 6601	*n* = 2849	*n* = 2508	*n* = 3885
**Deaths**	4590 (29.0)	1955 (29.6)	721 (25.3)	758 (30.2)	1156 (29.8)
**Deaths from NCD**	3378 (21.3)	1411 (21.4)	521 (18.3)	573 (22.8)	873 (22.5)
**Age, y**	45.0 ± 13.4	46.0 ± 13.5	43.2 ± 13.7	43.7 ± 12.4	45.3 ± 13.6
**Sex (*n*, % of Males)**	7754 (48.9)	2810 (42.6)	1280 (44.9)	1797 (71.7)	1867 (48.1)
**BMI**					
<25 kg/m^2^	8782 (55.4)	3737 (56.6)	1742 (61.1)	1218 (48.6)	2085 (53.7)
25–30 kg/m^2^	5326 (33.6)	2141 (32.4)	859 (30.2)	990 (39.5)	1336 (34.4)
v≤30 kg/m^2^	1720 (10.9)	716 (10.8)	245 (8.6)	297 (11.8)	462 (11.9)
Imputed	15 (0.1)	7 (0.1)	3 (0.1)	3 (0.1)	2 (0.1)
**Citizenship (*n*, % of Swiss)**	12,889 (81.4)	5358 (81.2)	2470 (86.7)	1908 (76.1)	3153 (81.2)
**Education**					
Mandatory	5458 (34.5)	2311 (35.0)	885 (31.1)	852 (34.0)	1410 (36.3)	
Upper secondary	7529 (47.5)	3034 (46.0)	1374 (48.2)	1248 (49.8)	1873 (48.2)	
Tertiary	2836 (17.9)	1250 (18.9)	585 (20.5)	404 (16.1)	597 (15.4)	
Imputed	20 (0.1)	6 (0.1)	5 (0.2)	4 (0.2)	5 (0.1)	
**Physical Activity**					
<1x/week	8682 (54.8)	3429 (51.9)	1437 (50.4)	1535 (61.2)	2281 (58.7)
1x/week	3477 (21.9)	1542 (23.4)	668 (23.4)	451 (18.0)	816 (21.0)
>1x/week	3441 (21.7)	1514 (22.9)	711 (25.0)	483 (19.3)	733 (18.9)
Imputed	243 (1.5)	116 (1.8)	33 (1.2)	39 (1.6)	55 (1.4)
**Smoking**					
Never	7476 (47.2)	3324 (50.4)	1480 (51.9)	809 (32.2)	1863 (48.0)
Former	2690 (17.0)	1154 (17.5)	426 (15.0)	513 (20.5)	597 (15.4)
Light	3214 (20.3)	1302 (19.7)	596 (20.9)	528 (21.1)	788 (20.3)
Heavy	2438 (15.4)	809 (12.3)	342 (12.0)	655 (26.1)	632 (16.3)
Imputed	25 (0.2)	12 (0.2)	5 (0.2)	3 (0.1)	5 (0.1)

Values are *n* (within group percentages) or mean ± standard deviation. The number of imputed values is only shown for variables with missing values. NCDs, noncommunicable diseases; BMI, body mass index; 1×/week: once a week.

**Table 2 nutrients-12-00833-t002:** The number of deaths, disability-adjusted life years, years of life lost, and years lost due to disability attributable to noncommunicable diseases in the NRP–MONICA cohort.

	Deaths	DALYs		YLL		YLD	
	*n*	sum	mean ± SD	sum	mean ± SD	sum	mean ± SD
All NCDs	3378	58,771	17.4 ± 8.2	42,348.1	12.5 ± 7.9	16,422.4	4.9 ± 2.6
Cancer ^1^	1593	32,573	20.4 ± 8.5	24,352.9	15.3 ± 8.2	8219.6	5.2 ± 2.6
CVD ^2^	1518	21,945	14.5 ± 6.8	14,842.2	9.8 ± 6.7	7102.7	4.7 ± 2.8
Respiratory ^3^	165	2597	15.7 ± 5.6	1841.3	11.2 ± 5.7	756.1	4.6 ± 1.7
Diabetes mellitus ^4^	51	629	12.3 ± 5.0	410.6	8.1 ± 5.1	218.2	4.3 ± 0.5
Chronic liver diseases ^5^	51	1027	20.1 ± 8.2	901.2	17.7 ± 8.2	125.8	2.5 ± 0.5

^1^ All neoplasms (ICD-8, 140–239; ICD-10, C00-D48); ^2^ diseases of the circulatory system (ICD-8, 390–459; ICD-10, I00-I99); ^3^ diseases of the respiratory system (ICD-8, 460–519; ICD-10, J30-J98); ^4^ diabetes mellitus (ICD-8, 249–250; ICD-10, E10-E14); ^5^ chronic liver diseases (ICD-8, 570–573; ICD-10, K70 and K74). NCDs, noncommunicable diseases; DALY, disability-adjusted life years; YLL, years of life lost; YLD, years lost due to disability; CVD, cardiovascular disease.

**Table 3 nutrients-12-00833-t003:** DALYs due to noncommunicable diseases and DALYs-dietary patterns association in the NRP–MONICA cohort.

	Overall	Prudent	Swiss Traditional	Western
*n*	15,843	6601	2849	2508
**Sum of DALYs**				
DALYs	58,771	23,697	9219	10,734
YLL	42,348	16,829	6624	7874
YLD	16,422	6868	2595	2860
**Mean ± SD of DALYs**				
DALYs	3.7 ± 8.1	3.6 ± 7.8	3.2 ± 7.8	4.3 ± 8.7
YLL	2.7 ± 6.3	2.5 ± 6.1	2.3 ± 6.1	3.1 ± 6.8
YLD	1.0 ± 2.3	1.0 ± 2.3	0.9 ± 2.2	1.1 ± 2.4
**DALYs-Dietary Patterns Association**				
Crude	-	Reference	−0.01 (−0.29; 0.26)	0.55 (0.24; 0.86)
Multi-adjusted	-	Reference	−0.01 (−0.28; 0.24)	0.29 (0.02; 0.56)

The crude model is adjusted for age and sex. The multi-adjusted model is adjusted for age, sex, nationality, education, BMI, physical activity, and smoking status. DALYs, disability-adjusted life years; YLL, years of life lost; YLD, years lost due to disability.

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
