# Peer review of "Combining Recent Nutritional Data with Prospective Cohorts to Quantify the Impact of Modern Dietary Patterns on Disability–Adjusted Life Years: A Feasibility Study"

_nutrients, 2020, doi:10.3390/nu12030833_

Round 1

Reviewer 1 Report

This is an interesting paper that has investigated the dietary patterns consumed. But I would suggest some changes  

In the introduction part must be developed. for example-please add literature related to this problem in your study area, why did you investigate ..etc.you need to talk scientifically.

The method is well laid out and provides a good overview.   supplementary really help to understand the results. I think this will really assist the reader in understanding the results, it will also assist future researchers to compare their results to yours.    Discussion part must be improved.example-you are talking limitation in the introduction part also. I suggest that you can discuss in the introduction part or discussion part.  

In conclusion please add future research feasibility.

Author Response

This is an interesting paper that has investigated the dietary patterns consumed. But I would suggest some changes 

Reply: We thank reviewer 1 for the evaluation of our work.

In the introduction part must be developed. for example-please add literature related to this problem in your study area, why did you investigate ..etc.you need to talk scientifically.

Reply: Our introduction now includes the following elements and additional references (5, 10-13) to better introduce the topic:

  • A justification of the use of DALYs for public health (l.36-42)
  • The advantage of calculating DALYs using cohort vs classic aggregated data (l.43-47)
  • The issue of calculating DALYs with outdated information from cohort entry (l.48-51)
  • How our methods is trying to solve this issue (l.51-61)

We removed some wording that was less scientific (ie, “sometimes 20 years ago” l. 49)

The method is well laid out and provides a good overview. supplementary really help to understand the results. I think this will really assist the reader in understanding the results, it will also assist future researchers to compare their results to yours.

Discussion part must be improved.example-you are talking limitation in the introduction part also. I suggest that you can discuss in the introduction part or discussion part. In conclusion please add future research feasibility.

Reply: In the introduction, we have discussed the limitations of previous methods (based on aggregated national data or cohort entry evaluation of diets, l43-51). In the discussion, we have listed and discussed the limitations of our novel proposed method (l.251-262 and 272-281 and 282-290). We have now added research feasibility and generalizability in the discussion (l.28-290 and 295-296).

Reviewer 2 Report

The purpose of this work was to quantify the impact of dietary patterns on Disability-Adjusted Life Years 17 (DALYs) from non-communicable diseases among a national cohort in Switzerland from two distinct data sets. The rationale for using more than one data set was to account for the fact that peoples’ dietary patterns tend to change over time and most cohort studies are limited from only relying on dietary data at a single time point.

Overall I think this is good work because it adds to literature on dietary patterns and DALYs by attempting to account for more recent dietary patterns and how they affect health outcomes. I just have one minor suggestion to improve the manuscript:

  • For paragraph 2.4, the assignment of participants to the different dietary patterns, it would be helpful to have a few sentences that give a general description of each one. For example, “the prudent dietary pattern is a dietary pattern generally based on a high intake of fruits and vegetables and minimal intake of red meats and alcohol.” Etc. Just so that the reader is oriented to each one.

Author Response

The purpose of this work was to quantify the impact of dietary patterns on Disability-Adjusted Life Years 17 (DALYs) from non-communicable diseases among a national cohort in Switzerland from two distinct data sets. The rationale for using more than one data set was to account for the fact that peoples’ dietary patterns tend to change over time and most cohort studies are limited from only relying on dietary data at a single time point.

Overall I think this is good work because it adds to literature on dietary patterns and DALYs by attempting to account for more recent dietary patterns and how they affect health outcomes. I just have one minor suggestion to improve the manuscript:

Reply: We thank reviewer 2 for the evaluation of our work.

For paragraph 2.4, the assignment of participants to the different dietary patterns, it would be helpful to have a few sentences that give a general description of each one. For example, “the prudent dietary pattern is a dietary pattern generally based on a high intake of fruits and vegetables and minimal intake of red meats and alcohol.” Etc. Just so that the reader is oriented to each one.

Reply: Thank you for this suggestion. We added this description at the end of paragraph 2.3 (l.118-121).